# A Label-Free and Data-Free Training Strategy for Vasculature Segmentation in serial sectioning OCT Data

**Etienne Chollet**                                    ECHOLLET@MGH.HARVARD.EDU
**Yaël Balbastre**                                    YBALBASTRE@MGH.HARVARD.EDU
**Caroline Magnain**                                    CMAGNAIN@MGH.HARVARD.EDU
**Bruce Fischl**\*                                    BFISCHL@MGH.HARVARD.EDU
**Hui Wang**\*                                    HWANG47@MGH.HARVARD.EDU
*A.A. Martinos Center for Biomedical Imaging, Massachusetts General Hospital, Boston, USA*

## Abstract

Serial sectioning Optical Coherence Tomography (sOCT) is a high-throughput, label free microscopic imaging technique that is becoming increasingly popular to study post-mortem neurovasculature. Quantitative analysis of the vasculature requires highly accurate segmentation; however, sOCT has low signal-to-noise-ratio and displays a wide range of contrasts and artifacts that depend on acquisition parameters. Furthermore, labeled data is scarce and extremely time consuming to generate. Here, we leverage synthetic datasets of vessels to train a deep learning segmentation model. We construct the vessels with semi-realistic splines that simulate the vascular geometry and compare our model with realistic vascular labels generated by constrained constructive optimization. Both approaches yield similar Dice scores, although with very different false positive and false negative rates. This method addresses the complexity inherent in OCT images and paves the way for more accurate and efficient analysis of neurovascular structures.

**Keywords:** Medical Imaging, Segmentation, Optical coherence tomography, Machine Learning, Data Synthesis.

## 1. Introduction

Vascular networks are central to brain function in health and disease, but mapping the human neurovasculature across scales, from mesoscale veins and arteries down to microscale capillaries, is an unmet challenge. Optical Coherence Tomography (OCT) (Huang et al., 1991) offers a wealth of information concerning vascular details. Combined with serial sectioning, sOCT reconstructs volumetric microvascular networks (Wang et al., 2018), but their automated segmentation is hindered by the presence of complex, structured, high-frequency noise. Traditional knowledge-based methods, that rely on hessian-based filters (Frangi et al., 1998; Yang et al., 2022), morphological operations (Zana and Klein, 2001), or region growing techniques (Dokládal et al., 1999) are highly sensitive to noise and fall short especially when vessels cross tissue boundaries, or when examining non-tubular morphologies. Conversely, convolutional neural networks (CNNs) have proven robust to these inconsistencies (Goni et al., 2022). However, the cost of producing labeled data results in low generalizability of the CNN models due to a lack of diverse training data. As an alternative, synthetic datasets have been increasingly leveraged in tasks of segmentation and

---

\* Joint senior authors

registration (Hoffmann et al., 2021; Billot et al., 2023). Synthetic vascular labels generated by constrained constructive optimization (CCO) (Schneider et al., 2012) have been used to train segmentation models for retinal OCT (Kreitner et al., 2024) and photoacoustic images (Sweeney et al., 2023), and to pretrain models targeting MR angiography and light-sheet microscopy (Paetzold et al., 2019; Tetteh et al., 2020). In contrast to realistic synthesis, Dey et al. (2024) recently showed that simple geometrical analogy can effectively yield a universal segmentation model in cells. Following this line of thought, we:

**1.** Propose a domain-randomized image generation strategy for sOCT volumes;

**2.** Propose an *unrealistic* spline-based synthesis engine that covers a wide range of vessel-like geometries, much larger than seen in practice;

**3.** Compare models trained with realistic CCO labels and unrealistic spline labels.

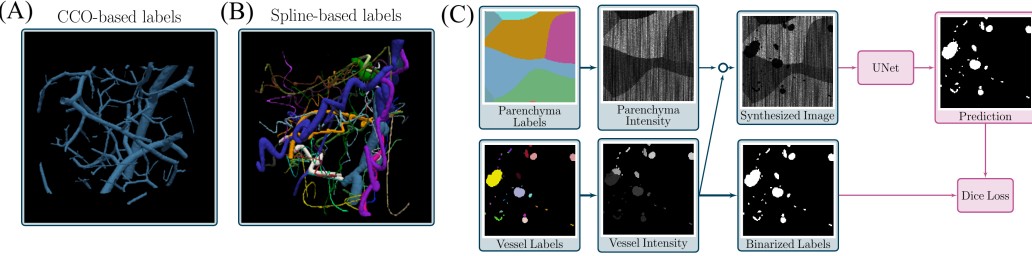

Figure 1: 3D renderings of CCO **(A)** and complex **(B)** labels, and **(C)** data synthesis pipeline for unsupervised training of UNet in vasculature segmentation task.

## 2. Methods

**Spline-based label synthesis.** We generate vascular trees by randomly drawing cubic splines within a 3D space. We initially sample straight lines before adding randomly jittered control points. Their number and the magnitude of the jitter define the spline tortuosity. Branching points are then sampled to serve as the endpoints of a second set of splines, and the procedure is repeated recursively. The spline radius is allowed to vary along its length and is also encoded by cubic splines. The change in radius across levels in the recursion is randomly sampled. Finally, splines are rasterized using a distance transform computed by Gauss-Newton optimization. We generated two different datasets, labeled *complex* and *simple*, that respectively used wide and peaked distribution for all hyper-parameters.

**CCO-based label synthesis.** We used the synthetic labels released in Tetteh et al. (2020), which were generated using the procedure described in Schneider et al. (2012).

**Domain-randomized OCT synthesis.** Vessel-specific and parenchyma-specific textures are generated by sampling two different random label maps, following the procedure from Hoffmann et al. (2021). A random intensity, between 0 and 1, is assigned to each label, and the parenchyma and vessels are fused *via* voxel-wise multiplication. Thus, a vessel's intensity is always lower than its surrounding tissue. (*i.e.*, "dark vessels"). This is crucial in OCT, as refractive effects can generate an artifactual bright shadow above the true vessel. Finally, random slab-wise intensity non-uniformities (that cover absorption, depth of focus,

stitching artifacts between slices) and speckle noise are applied.

**Model and training.** A U-Net (Ronneberger et al., 2015) with residual blocks (He et al., 2016) (number of parameters: $7.4 \cdot 10^6$) and Tilborghs et al. (2022)'s Dice loss was trained with Adam (learning rate: $10^{-2}$ with 2,000 warmup steps and 20,000 cooldown steps) for 100,000 steps. Each model was trained four times with different initial random weights. All training patches were of size $128 \times 128 \times 128$.

**Validation.** All models were evaluated on a $301 \times 301 \times 301$ voxel OCT volume manually labeled by an expert. A sine-weighted moving window was used at inference time.

A schematic of the complete pipeline is presented in figure 1.

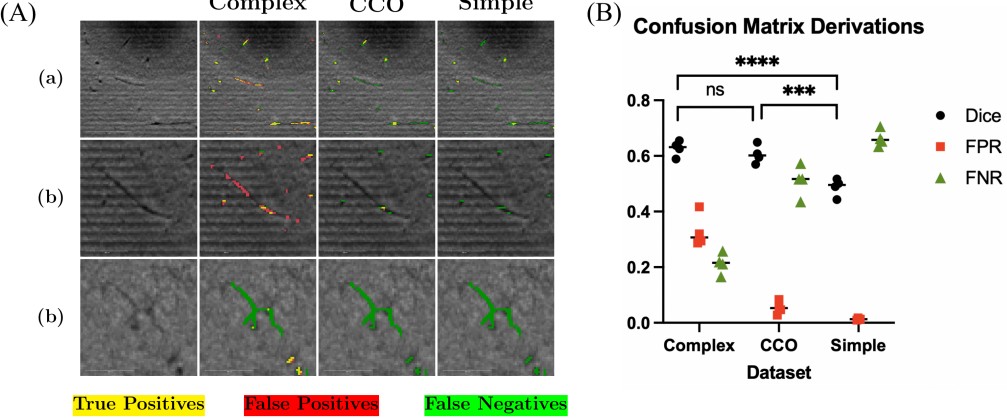

Figure 2: **(A)**: Qualitative comparison of model performance. **(B)**: Dice score, false positive rate (FPR), and false negative rate (FNR) for models (n=4) trained on complex, CCO, and simple datasets. ANOVA analysis was used to delineate statistical significance (****: $P < 0.0001$, ***: $P < 0.001$)

## 3. Results & Discussion

Figure 2 presents a comparison of the mean dice scores, false positive rates (FPR), and false negative rates (FNR) for each synthesis pathway. A two-way ANOVA shows that while vessel labels with simple parameters hurt network performance, networks trained on complex labels (dice=0.627) and CCO labels (dice=0.605) were similarly accurate. However, these two networks have drastically different error patterns, with the CCO models tending to undersegment (FPR=0.055, FNR=0.511) and the complex models tending to oversegment (FPR=0.330, FNR=0.213). A qualitative assessment of the segmentation shows that many of the false positives are in fact small but real vessels that were missed by the expert (see *e.g.* figure 2(A.b)). This is the foundation of the benefit of label-conditioned image generation, which guarantees a precise alignment of labels and image attributes. To train a segmentation model, we have shown that it is unnecessary to construct vessels according to rigid physical models. This discovery highlights the possibility of using synthetic data to build accurate segmentation models.

## Acknowledgments

Much of the computation resources required for this research was performed on computational hardware generously provided by the Massachusetts Life Sciences Center.

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
