# OpenReview forum: "A Label-Free and Data-Free Training Strategy for Vasculature Segmentation in serial sectioning OCT Data"
_MIDL.io/2024/Short_Papers — MIDL 2024 Short Papers_

### Official Review · Reviewer_g66B · 2024-04-24

**Confidence:** 5
**Final Rating:** 5

**Review:**

The paper introduces a novel approach for blood vessel segmentation in OCT images by using synthetic data with varying levels of realism for training a segmentation model, addressing the challenge of limited labeled data in medical imaging tasks. The work has the potential to significantly improve neurovascular structure analysis. The paper provides a clear evaluation by comparing different training strategies and their impact on segmentation accuracy and error patterns.

A few minor points concerning :
- further testing on larger and more varied datasets is required to validate results.
- the generation of synthetic data may require additional domain expertise. It would be worth investigating the possibility of refining the model trained on complex data to minimize false positives while keeping false negatives low.

---

### Decision · Program_Chairs · 2024-04-26

Accept